# Effects of Hyperbaric Oxygen Therapy on Serum Adhesion Molecules, and Serum Oxidative Stress in Patients with Acute Traumatic Brain Injury

**DOI:** 10.3390/jpm11100985

**Published:** 2021-09-29

**Authors:** Hung-Chen Wang, Pei-Ming Wang, Yu-Tsai Lin, Nai-Wen Tsai, Yun-Ru Lai, Chia-Te Kung, Chih-Min Su, Cheng-Hsien Lu

**Affiliations:** 1Departments of Neurosurgery, Kaohsiung Chang Gung Memorial Hospital and Chang Gung University College of Medicine, Kaohsiung 833401, Taiwan; m82whc@yahoo.com.tw; 2Departments of Family Medicine, Kaohsiung Chang Gung Memorial Hospital and Chang Gung University College of Medicine, Kaohsiung 833401, Taiwan; wangpeming@yahoo.com.tw; 3Departments of Otolaryngology, Kaohsiung Chang Gung Memorial Hospital and Chang Gung University College of Medicine, Kaohsiung 833401, Taiwan; whc8131977@gmail.com; 4Graduate Institute of Clinical Medical Sciences, College of Medicine, Chang Gung University, Taoyuan City 33302, Taiwan; 5Departments of Neurology, Kaohsiung Chang Gung Memorial Hospital and Chang Gung University College of Medicine, Kaohsiung 833401, Taiwan; tsainw@yahoo.com.tw (N.-W.T.); yunru.lai@gmail.com (Y.-R.L.); 6Departments of Emergency Medicine, Kaohsiung Chang Gung Memorial Hospital and Chang Gung University College of Medicine, Kaohsiung 833401, Taiwan; kungchiate@gmail.com (C.-T.K.); mitosu@gmail.com (C.-M.S.); 7Department of Biological Science, National Sun Yat-sen University, Kaohsiung 80424, Taiwan; 8Department of Neurology, Xiamen Chang Gung Memorial Hospital, Xiamen 361126, China

**Keywords:** traumatic brain injury, hyperbaric oxygen therapy, oxidative stress, adhesion molecules

## Abstract

Background: Serum concentrations of adhesion molecules and oxidative stress is thought to participate in the pathobiology of secondary brain injury after acute traumatic brain injury (TBI). We aimed to study the hypothesis that hyperbaric oxygen therapy (HBOT) both improves the adhesion molecules levels and antioxidant capacity. Methods: Thirty blood samples from ten patients after acute TBI were obtained after injury and before and after HBOT. Four patients received early HBOT started two weeks after injury, four patients received late HBOT started ten weeks after injury and two patients did not receive HBOT and served as control in this study. The HBOT patients received total 30 times HBOT in six weeks period. Results: Those serum biomarkers in patients with TBI had not significantly difference in glutathione (GSH), thiobarbituric acid reactive substances (TBARS), soluble intercellular cell adhesion-molecule-1 (sICAM-1) and soluble vascular cell adhesion molecule-1 (sVCAM-1) concentrations on admission between early HBOT, late HBOT, and control group (*p* = 0.916, *p* = 0.98, *p* = 0.306, and *p* = 0.548, respectively). Serum GSH levels were higher at 10 weeks after injury in the early HBOT group than in the late HBOT group and control group (mean, 1.40 μmol/L, 1.16 μmol/L, and 1.05 μmol/L, respectively). Then the serum GSH level was increased at 18 weeks after injury in the late HBOT group (mean, 1.49 μmol/L). However, there was only statistically significant difference at Weeks 18 (*p* = 0.916, *p* = 0.463, and *p* = 0.006, at Week 2, Week 10, and Week 18, respectively). Serum TBARS levels were decreased at 10 weeks after injury in the early HBOT group than in the late HBOT group and control group (mean, 11.21 μmol/L, 17.23 μmol/L, and 17.14 μmol/L, respectively). Then the serum TBARS level was decreased at 18 weeks after injury in the late HBOT group (mean, 12.06 μmol/L). There was statistically significant difference after HBOT (*p* = 0.98, *p* = 0.007, and *p* = 0.018, at Week 2, Week 10, and Week 18, respectively). There was no statistically significant difference between the three groups on sICAM-1 and sVCAM-1 levels from Week 2 to Week 18. Conclusions: HBOT can improve serum oxidative stress in patients after TBI. These molecules may be added as evaluation markers in clinical practice. Perhaps in the future it may also become part of the treatment of patients after acute traumatic brain injury. Further large-scale study may be warrant.

## 1. Introduction

Acute traumatic brain injury (TBI) is a disease process with an initial injury, involves inflammatory pathways activation, and contributes to continuing biochemical and cellular changes over time [1,2]. The brain has the highest oxygen metabolism rate among the organs of the body [3]. This high rate of oxygen metabolism leads to the possibility of excessive reactive oxidative stress (ROS) production. A large amount of ROS causes oxidative stress have been suggested to play a crucial role in the pathology of TBI, and several cell characteristics of the brain indicate that it is highly sensitive to oxidative stress. An imbalance of ROS generation and the oxidative metabolism which leads to mitochondrial dysfunctions and apoptosis [4], and that apoptotic factor in different brain regions of rats might contribute to neurotoxicity [5]. Brain tissue contains a lot of unsaturated fatty acids and iron, which can be metabolized by oxygen free radicals [6,7]. In scavenging oxygen free radicals, the brain has only 10% of the catalase activity in the liver [7]. Thus, compared with other body organs, the brain’s defense system against oxidative stress may be insufficient. Evidence from animal TBI models shows that ROS production increases immediately within a few minutes after injury [8,9,10]. Previous reports showed that excess oxidative stress can affect energy metabolism [4], block the electron transport chain of mitochondria and damage to neurocytes and cerebrovascular endothelium [11] causing neurotoxicity such as apoptosis [12,13,14]. Scavenging of ROS after TBI can reduce brain damage in rat [15].

There is growing evidence show that TBI can trigger systemic inflammation and coagulation cascades, leading to secondary neuropathological sequelae [16,17]. Vascular endothelium provides a key interface for the host’s inflammation and coagulation response to injury. These responses are related to the activation of resident and peripheral macrophages [18,19,20,21], and the activation of inflammatory cytokines, and adhesion molecules [22,23,24,25]. Inflammation cells marginalized along the vessel wall is mediated by adhesion to endothelial ligands for soluble intercellular (sICAM-1) and soluble vascular cell adhesion molecules (sVCAM-1), which allows leukocyte migration into the injured tissues [26]. Several clinical studies have demonstrated significant elevation of adhesion molecules in acute brain injury [27,28].

It is well documented that hyperbaric oxygen therapy (HBOT) has neuroprotective effects against damaged brain tissue [29,30,31,32,33,34]. Although HBOT had no effect to treat cognitive, or fine motor deficits associated in patients with mild TBI and post-concussive symptoms [32,34]. Some animal studies revealed that HBOT had the beneficial effect of on the injured brain and improved cognitive function [35,36,37,38]. Till now, little is known about the effects of HBOT on time course of oxidative stress concentration and soluble endothelial adhesion molecules changes in acute TBI patients. This prospective study aimed to evaluate the effects of HBOT on serial oxidative stress and adhesion molecular levels in patients with acute TBI.

## 2. Materials and Methods

### 2.1. Patients

The diagnosis of acute TBI is confirmed by medical history and computed tomography (CT) scan of the brain. Patients were excluded if they: (1) were taking anti-platelet or anti-coagulant; (2) penetrating head injury; (3) central nervous system infection; (4) had major systemic diseases like malignant tumor, end-stage renal disease, liver cirrhosis, congestive heart failure.

This study included mild and moderate (GCS ≦ 9) traumatic brain injuries. The age of the patients was range 18–65 years. Patients will be included in this study only if they have obtained the fully informed written consent of the patients or their family members and meet the above diagnostic criteria. Patients were assigned to different groups by principal investigator based on patient’s clinical condition. If patient was suitable to do HBOT, he was assigned by randomizing. If patient’s condition was not suitable to do HBOT in acute phase, he was assigned to late HBOT group, and if patient hesitated to do HBOT, he was assigned to No HBOT group. The informed consent will be approved by the Ethics Committee of our hospital. A total of ten adult patients were included in the study.

### 2.2. Hyperbaric Oxygen Therapy (HBOT)

During HBOT, patients were placed in a chamber and were pressurized with air to 2.5 atmospheres absolute (ATA) during 15 min, then were supplied 100% oxygen for 25 min, followed by a 5-min air break. Repeats the cycle once, and then gives 100% oxygen for 10 min, and then the chamber is depressurized to 1 ATA with 100% oxygen over 15 min. The total treatment time was 100 min.

### 2.3. Blood Sampling, Assessment of Oxidative Stress and Adhesion Molecules

Blood samples were collected from 10 acute TBI patients within 2 weeks after the injury, between the 8th and 10th weeks after the injury, and between the 16th and 18th weeks after the injury. Blood samples were collected into Vacutainer SST tubes (BD, Franklin Lakes, NJ, USA) by venipuncture. The blood was allowed to clot in room temperature for a minimum of 30 min. All samples were collected after centrifugation at 3000 rpm for 10 min at 4 °C, isolated, and immediately stored in multiple aliquots at −80 °C. The flow chart of patients recruited and treatments was on Figure 1.

#### 2.3.1. Serum Glutathione (GSH) Levels

The serum total protein glutathione is estimated by the direct reaction of glutathione with 5,5-dithiobis-2-nitrobenzoic acid (DTNB) to form 5-thio-2-nitrobenzoic acid (TNB). The amount of glutathione in the sample is calculated from the absorbance using the extinction coefficient of TNB (A412 = 13,600 M^−1^cm^−1^). In this study, we used the GSH Assay Kit (catalog 703002; Cayman Chemical, Michigan, MI, USA).

#### 2.3.2. Serum Thiobarbituric Acid-Reactive Substances (TBARS) Levels

In this study, the TBARS detection kit was used for rapid photometric detection of thiobarbituric acid-malonaldehyde (TBA-MDA) adducts at 532 nm, as described by the manufacturer (catalog 10009055; Cayman Chemical, Michigan, MI, USA). The sample value is calculated from a linear calibration curve using pure MDA samples (range: 0–50 μmol/L).

#### 2.3.3. Serum Soluble Intercellular Cell Adhesion-Molecule-1 (sICAM-1) Levels, and Serum Soluble Vascular Cell Adhesion Molecule-1 (sVCAM-1) Levels

The serum levels of sICAM-1 and sVCAM-1 were evaluated by a commercially available ELISA (R&D Systems, Minneapolis, MN, USA). The absorbance is directly proportional to the concentration of antigen present. A set of antigen standards is used to draw a standard curve between absorbance and antigen concentration, from which the antigen concentration in the unknown is calculated.

### 2.4. Clinical Manifestations

The patients were also divided into three groups according HBOT: early HBOT group (received HBOT between 2 weeks and 8 weeks after injury), late HBOT (received HBOT between 10 weeks and 16 weeks after injury), and control group (did not receive HBOT).

Patients receive regularly monitor the Glasgow Coma Scale (GCS) score, and vital sign, keep fluid balance, and normalize laboratory parameters. Using the objective Injury Severity Score (ISS) on admission to calculate the total degree of injury [39]. All patients underwent brain CT scans in the emergency room. If clinical deterioration, such as acute onset of focal neurological deficits, seizures, status epilepticus, and progressive disturbance of consciousness, repetitive brain CT scans or/and magnetic resonance imaging (MRI) and as a routine after neurosurgery.

### 2.5. Statistics

Categorical data were analyzed by chi-square test or Fisher’s exact test. Use Student’s test or ANOVA to analyze continuous data. The data is the mean ± standard derivation (SD) for normally distributed data. The SPSS 22.0 (SPSS Inc., Chicago, IL, USA) software was used for all statistical analyses.

## 3. Results

### 3.1. Baseline Characteristics of the Study Patients

According to the baseline characteristics of acute TBI cases (Table 1), the 10 acute TBI patients included 5 males (age range 21–48 years; median age 36 years) and 5 females (age range 27–70 years; average age, 57 years old). According to the GCS score at admission, 6 cases were mild, 1 case was moderate, and 3 cases were severe. The ISS on admission was 20. Three patients had moderate injuries (ISS 9–15), five had severe injuries (ISS 16–24), and two had very serious injuries (ISS > 24). Four patients (40%) underwent neurosurgery within 24 h after TBI, of which three underwent craniotomy and one underwent intracranial pressure monitoring. The mean values of GCS score and ISS for patients undergoing neurosurgical treatment at admission were 11.5 and 24.8, respectively. The most common CT findings at the time of presentation were parenchymal contusion hemorrhage (50%), traumatic SAH (40%) and subdural hemorrhage (40%).

Comparison of these three groups, early HBOT, late HBOT and no HBOT, at admission, there were not significantly difference at baseline data, except worse GCS in the late HBOT group (Table 1). Those serum biomarkers in patients with TBI had not significantly difference in Glutathione, TBARS, sICAM-1 and sVCAM-1 concentrations on admission (*p* = 0.916, *p* = 0.98, *p* = 0.306, and *p* = 0.548, respectively) (Table 1).

### 3.2. The Time Course of Oxidative Stress and Serum Adhesion Molecules Concentration Changes

The time course of oxidative stress and serum adhesion molecules concentration changes in acute TBI patients with early and late HBOT was compared. Serum GSH levels were higher at 10 weeks after injury in the early HBOT group than in the late HBOT group and control group (mean, 1.40 μmol/L, 1.16 μmol/L, and 1.05 μmol/L, respectively). Then the serum GSH level was increased at 18 weeks after injury in the late HBOT group (mean, 1.49 μmol/L). However, there was no statistically significant difference at Weeks 18 (*p* = 0.916, *p* = 0.463, and *p* = 0.060, at Week 2, Week 10, and Week 18, respectively) (Figure 2A). Serum TBARS levels were decreased at 10 weeks after injury in the early HBOT group than in the late HBOT group and control group (mean, 11.21 μmol/L, 17.23 μmol/L, and 17.14 μmol/L, respectively). Then the serum TBARS level was decreased at 18 weeks after injury in the late HBOT group (mean, 12.06 μmol/L). There was statistically significant difference after HBOT (*p* = 0.98, *p* = 0.007, and *p* = 0.018, at Week 2, Week 10, and Week 18, respectively) (Figure 2B). Post-hoc test showed early HBOT was statistically significant difference on late HBOT (*p* = 0.013) and control group (*p* = 0.021) at Week 10, and control group was statistically significant difference on late HBOT (*p* = 0.035) and early HBOT (*p* = 0.031) at Week 18. There was no statistically significant difference between the three groups on sICAM-1 and sVCAM-1 levels from Week 2 to Week 18 (Figure 3A,B).

By GOS upon discharge of the 10 acute TBI patients, six had no disability, four had moderated disability. There was no statistically significant difference between the three groups on Week 10 and Week 18 following-up (*p*= 0.193 and *p* = 0.193, at Week 10 and Week 18, respectively).

## 4. Discussion

This study examined the correlation between HBOT and markers of oxidative stress and endothelial cell activation after acute TBI. There are several main findings. First, compared with the control group, the oxidative stress concentration of GSH and TBARS in the HBOT group was significantly increased after HBOT. Secondly, plasma sICAM-1 and sVCAM-1 levels did not change significantly after HBOT. Third, comparing the treatment results of acute TBI cases based on GOS, there was no significant difference in HBOT between the three groups.

### 4.1. Oxidative Stress in TBI Patients

GSH is an endogenous antioxidant that can protect endothelial function and increase the bioavailability of nitric oxide in neurovascular units after acute injury. The brain GSH level after TBI measured 24 h after cortical contusion in rats was significantly lower than that of control brains [40,41]. Compared to the control group, serum GSH level was increased at 10 weeks in the early HBOT group and persisted to Week 18. This meant that GSH was triggered and persisted for at least eight weeks after HBOT. The serum GSH level was increased at 18 weeks in the late HBOT group. In contrast, the serum GSH level was no significant changed in control (TBI but no HBOT) group. HBOT did increase serum GSH level, however, there was only statistically significant difference at Weeks 18 between these three groups (*p* = 0.916, *p* = 0.463, and *p* = 0.006, at Week 2, Week 10, and Week 18, respectively) (Figure 2A). However, in Bayir et al. study showed that compared with the control group, GSH levels increased on day 1, and then decreased in severe TBI [42]. This posits that GSH is triggered in the early stage of TBI, at least on Day 1, and then decreases thereafter.

In the present study, HBOT did increase serum GSH level, however, there was only statistically significant difference at Weeks 18 between these three groups (*p* = 0.916, *p* = 0.463, and *p* = 0.006, at Week 2, Week 10, and Week 18, respectively) (Figure 2A). Bayir et al. another study of 13 severe pediatric TBI patients with therapeutic hypothermia showed that hypothermia can preserve CSF antioxidant reserves [43]. The study also showed that the level of GSH in CSF was negatively correlated with the patient’s body temperature at the time of sampling after injury. Different results of Bayir et al. 2002, and present study may be due to different disease severity (severe TBI in Bayir’s study, and mainly mild TBI in this study), different samples (cerebrospinal fluid and plasma), or different patient ages (children and aldult).

Serum TBARS levels was significant decreased at 10 weeks in the early HBOT group and persisted to Week 18. This meant that TBARS was supressed and persisted for at least eight weeks after HBOT. Then the serum TBARS level was decreased at 18 weeks after injury in the late HBOT group. There was statistically significant difference after HBOT (*p* = 0.98, *p* = 0.007, and *p* = 0.018, at Week 2, Week 10, and Week 18, respectively) (Figure 2B).

In the study of Kasprzak et al., compared with the control group, during the 10-day follow-up period, the CSF erythrocyte TBARS concentration of brain contusion patients increased significantly [44]. The highest CSF TBARS concentration was observed in 5 patients who died 2, 7, or 8 days after head injury. The results of the current study demonstrates that the HBOT group has significantly improved oxidative stress concentrations of GSH and TBARS after HBOT.

### 4.2. Adhesion Molecules in TBI Patients

Soluble adhesion molecules has been proposed to inhibit the ongoing immune inflammatory response by competitive binding or by inducing the response of ligand-bearing cells [45]. The endothelium promotes inflammatory cells to enter the brain parenchyma, which depends on the expression and regulation of paired adhesion molecules presenting on infiltrating leukocytes and cerebral vascular endothelial cells [46,47].

The level of serum sICAM-1 in patients with multiple injuries is elevated, which is related to the severity of organs dysfunction [48]. Adhesion molecules play an important role in the process of inflammation and may play different roles in different conditions. One animal study using ICAM-1-deficient mutant mice has shown that ICAM-1 does not cause the pathogenesis of TBI [49]. However, some studies have shown that circulating sICAM-1 and sVCAM-1 levels are elevated in patients with acute TBI, and sVCAM-1 levels are significantly elevated in TBI patients with poor outcome [27].

### 4.3. Study Limitations

The current study has several limitations. First, serum GSH and TBARS levels are only part of the oxidative stress expressed after TBI. Therefore, the change levels of these markers may not necessarily reflect their actual pathophysiological functions. Second, different medical treatments by different doctors during HBOT may be a confounding factor. Third, the sample size is small. This may lead to potential deviations in statistical analysis or accidental statistical discoveries. Last, the Glasgow Outcome Score (GOS) is a very blunt instrument, and unlikely to show a functional effect of treatment, particularly in such a small sample.

## 5. Conclusions

The HBOT group has significantly improved oxidative stress concentrations of GSH and TBARS after HBOT and persisted for at least eight weeks. However, HBOT did not improve plasma sICAM-1 and sVCAM-1 levels. In clinical practice, the measurement of such molecules can be added to the trauma parameter as evaluation markers. Serum oxidative stress concentration may be a potential therapeutic target for patients with TBI. Nevertheless, a large-scale study with a larger research population should be conducted to verify the results.

## Figures and Tables

**Figure 1 jpm-11-00985-f001:**
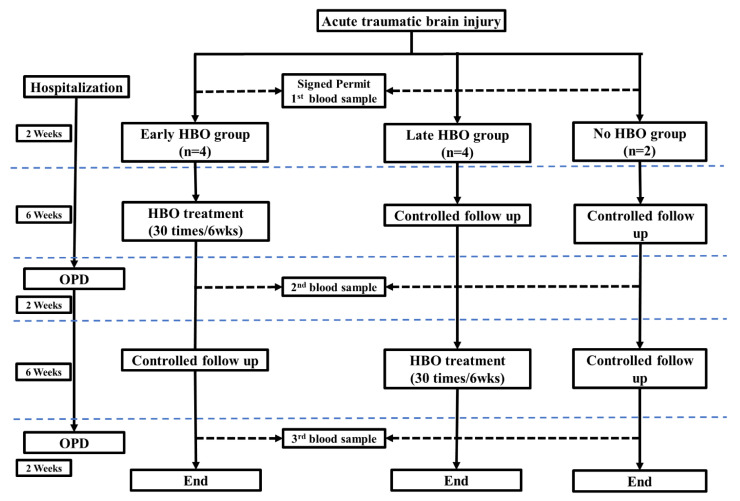
The flow chart of patients recruited and treatments.

**Figure 2 jpm-11-00985-f002:**
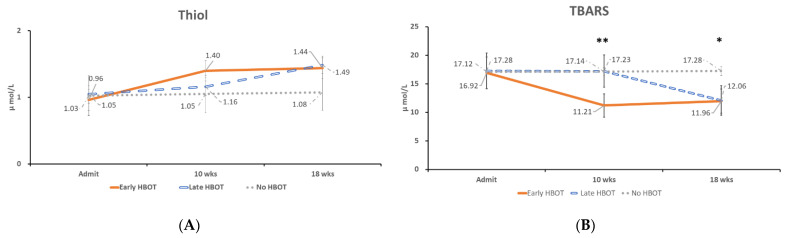
The time course of serum oxidative stress concentration changes in (**A**) GSH, and (**B**) TBARS at Week 2, Week 10, and Week 18 in patients with acute TBI who received early HBOT, late HBOT and control group. * *p* < 0.05, ** *p* < 0.01, between early HBOT group, late HBOT group and control group, by ANOVA.

**Figure 3 jpm-11-00985-f003:**
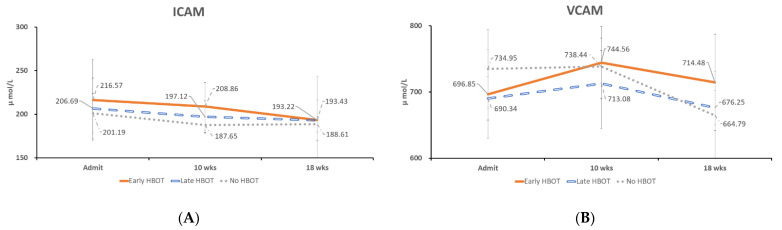
The time course of serum adhesion molecular concentration changes in (**A**) sICAM-1, and (**B**) sVCAM-1 at Week 2, Week 10, and Week 18 in patients with acute TBI who received early HBOT, late HBOT and control group.

**Table 1 jpm-11-00985-t001:** Demographic Data in Patients with Acute Traumatic Brain Injury at Admission.

	Early HBOT (*N* = 4)	Late HBOT (*N* = 4)	No HBOT (*N* = 2)	*p*-Value
Age (y)	53 ± 10.3	41 ± 10.9	44 ± 4	0.708
BMI	26.4 ± 1.8	25.2 ± 1.1	22.6 ± 1.2	0.346
Male	2	2	1	0.223
Underlying diseases				
Diabetes mellitus	2	0	0	0.153
Hypertension	2	1	0	0.435
Coronary artery disease	0	0	0	NA
Alcoholism	0	0	0	NA
Smoking	0	0	0	NA
Clinical feature at presentation				
Brief unconsciousness	1	3	1	0.368
Motor deficits	1	3	0	0.153
Posttraumatic amnesia	1	3	0	0.153
GCS at presentation	15 ± 0	8.5 ± 0.9	15 ± 0	≤0.001
Injury Severity Score at presentation	19.3 ± 2.1	19.8 ± 3.7	22.5 ± 11.5	0.9
Laboratory data at presentation				
WBC (×10^3^/mL)	11.4 ± 1.9	17.3 ± 2.6	15.1 ± 1.8	0.223
Hemoglobin (gm/dl)	12.8 ± 1.0	12.1 ± 1.2	13.9 ± 0.1	0.611
Hematocrit	38.9 ± 3.1	36.9 ± 3.7	42.0 ± 0.5	0.655
Platelet counts (×10^3^/mL)	230.2 ± 14.9	303.8 ± 37.5	228.5 ± 10.5	0.167
Prothrombin Time (PT)	10.5 ± 0.38	10.8 ± 0.43	11.1 ± 0.15	0.701
Activated partial thromboplastin time (APTT)	23.7 ± 0.5	25.7 ± 1.9	26.4 ± 0.7	0.432
Brain Imagies Findings at presentation				
Parenchymal contusion hemorrhage	2	3	0	0.223
Epidural hemorrhage	2	1	1	0.732
Subdural hemorrhage	2	2	0	0.435
Traumatic subarachnoid hemorrhage	1	1	2	0.153
Depressed skull fracture	0	0	0	NA
Pneumocranium	0	0	0	NA
Neurosurgical intervention	1	2	1	0.732
Oxidative Stress at presentation				
Glutathione (μmol/L)	0.96 ± 0.15	1.05 ± 0.16	1.03 ± 0.17	0.916
TBARS (μmol/L)	16.92 ± 1.40	17.28 ± 1.55	17.12 ± 0.19	0.98
Soluble intercellular adhesion molecule at presentation				
sICAM-1 (ng/mL)	216 ± 15	246 ± 17	201 ± 13	0.306
sVCAM-1 (ng/mL)	696 ± 33	690 ± 16	732 ± 34	0.548
Acute neurosurgical complications				
Newly onset of neurological deficit	0	0	0	NA
Deterioration of consciousness	0	0	0	NA
Posttraumatic seizure	0	0	0	NA
GOS at 10 wks follow-up	5 ± 0	4.5 ± 0.29	5 ± 0	0.193
GOS at 18 wks follow-up	5 ± 0	4.5 ± 0.29	5 ± 0	0.193
Days of ICU stay	2.8 ± 1.4	3.8 ± 1.3	3 ± 0	0.848
Days of Hospitalization	11.8 ± 1.3	25 ± 5	9 ± 4	0.134

GCS, Glasgow coma scale; SD, standard derivation; TBARS, thiobarbituric acid-reactive substances; sICAM-1, soluble intercellular adhesion molecule; sVCAM-1, soluble vascular cell adhesion molecule GOS, Glasgow outcome scale; ICU, intensive care unit. Data are presented either as absolute numbers or mean ± standard derivation (SD). Statistical significance was set at a level of *p* = 0.05. Statistical variance between groups was assessed by the Fisher’s exact test for discrete variables and by Student’s *t* test or ANOVA test for continuous variables.

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
