# Peer review of "Effects of Hyperbaric Oxygen Therapy on Serum Adhesion Molecules, and Serum Oxidative Stress in Patients with Acute Traumatic Brain Injury"

_jpm, 2021, doi:10.3390/jpm11100985_

Round 1

Reviewer 1 Report

Comments for Authors:

The article is intresting and provides the knowledge that  hyperbaric oxygen therapy can improve serum oxidative stress in patients after TBI. I would ask the authors:

  1. Please use accurate scientific terminology instead of “ tested” in the abstrct.
  2. “Oxidative stress has been suggested to play a crucial role in the pathology of TBI” through which machnism oxidative stress play a key role? If explain breifly, will improve the quality of the manuscript.
  3. Include the word “ production” like This, (High rate of oxygen metabolism leads to the possibility of excessive reactive oxidative stress (ROS) production.
  4. “Several previous studies have shown that hyperbaric oxygen therapy (HBOT) elevated levels of dissolved oxygen can have multiple repair effects on damaged brain tissue” review and change the sentence by simple one.
  5. “during 15 min and were supplied 100% oxygen for 25 mins, followed by a 5-min air break. The cycle was repeated once, and then 100% oxygen was used for 10 minutes” review the sentence i.e min vs mins vs minutes.
  6. “NO” write the full form. Line # 10 in disscusion.
  7. “ Different results of Bayir et al ” add full refrence or should write like this (Bayir et al.2002,
  8. In the INTRODUCTION of 3rd paragraph of 6th line please change TO DATE to TILL NOW.
  9. MATERIALS AND METHODS of 2nd paragraph in the 2nd line please put RANGE before the 18-65 years.
  10. 5 paragraph in the 1st line please change ARE to WERE. And also delete T after STUDENT,S.
  11. In the 3.RESULT of 2nd paragraph of 1st line please change COMPARED TO COMPARISON.

Reviewer 2 Report

The manuscript describes a comparison of post-TBI patients exposed to hyperbaric oxygen treatment at various times after TBI. The authors find that there are changes in serum markers of oxidative stress including TBARS and glutathione concentrations, without significant changes to functional outcomes (as measured using Glasgow Outcome Scales). The results are interesting in that they show persistent changes for weeks after hyperbaric oxygen, and may shed some light on potential mechanisms that may explain effects of hyperbaric oxygen treatment. The manuscript overall does suffer from a lack of detail, particularly regarding methods and statistical analysis, and would benefit greatly from clarifications. In addition, the English usage is sometimes a little difficult to understand.

Specific comments:

- Introduction, last sentence of first paragraph: the paper cited here (reference 9) does not report and neurologic markers, so does not support the assertion that ROS scavenging can improve function after TBI.

- Introduction, last sentence: This suggests the studies were undertaken to evaluate the relationship between oxidative stress markers and outcomes; there does not appear to be much discussion of this in the paper though.

- Methods, patients: How were patients recruited? How were they assigned to groups, and who made the decision on HBOT treatment? When in the post-injury course were they enrolled in the study?

- Methods, HBOT: Was HBOT performed only once on each patient who received it? It would also be useful to know what the spread in timing of HBOT is. i.e., when did each patient get HBOT? My impression of the manuscript is that there was probably a fairly continuous distribution of times post-injury that were later divided into early and late, but it is not clear.

- Methods, Blood sampling: more detail is needed about how blood samples were obtained, and especially how samples were handled after collection.

- Methods, GSH levels: how was this assay done? For other measurements it appears that commercial kits were used, but it is not clear for this one.

- Methods, Clinical manifestations: Again, how were patients divided into groups? This can be put into either the section on patients or the one on HBOT, per the authors’ preference. It is not clear what the other reported measurements were used for, such as observation, GCS, ECG, vital signs, and imaging, since results of these were not reported anywhere in the paper.

- Methods, statistics: statistical methods appear to be appropriate for the data analyzed. For data analyzed by ANOVA, though, it is not clear whether a post-hoc analysis was used, or if the statistics reported are just for the omnibus ANOVA test. This applies to the results reported in Fig. 1 and Fig 2.

- Figure 1 legend: It is not clear what is different by the asterisks; ** is not defined, and for points labeled with *, it is not clear which groups are significantly different from each other. For example, on Fig 1A, are both early and late HBOT different from controls?

- Discussion, section 4.1: the authors refer to “oxidative stress concentration of GSH and TBARS.” It is not completely clear to me what this means, but it does appear that the authors are using GSH and TBARS as markers of systemic oxidative stress. There needs to be more discussion of the justification of these as markers of oxidative stress (I agree that these are valid measures, but there is no attempt to justify why these measures were chosen for this study).

- Discussion, section 4.2, last paragraph: Does the first sentence refer to the changes seen in this study? There was no baseline measure taken or cited to support that the levels seen are elevated. Or does this sentence refer to the data cited above? In that case, this should be made clearer. In either case, the fact that sICAM and sVCAM are elevated does not prove that “the activation of adhesion molecules played a pathophysiological role in the acute phase after TBI” as stated in this section. If the authors wish to make this assertion about the pathophysiologic role of adhesion molecules, more discussion would be needed; however, I do not think this necessarily has to be done, since there wasn’t a change in these due to the HBOT treatment.

- Discussion, section 4.3: Another significant limitation I would cite is that the Glasgow Outcome Score (GOS) is a very blunt instrument, and unlikely to show a functional effect of treatment, particularly in such a small sample.

Minor points:

- Figures 1 and 2: the font size for the figure labels is very small, and hard to read.

- Figure 2 legend: * is defined as p < 0.05, but there is no asterisk on the graph.

- At the end, Data Availability Statement: nothing appears to have been added here.
